# Neutral Polysaccharide from the Leaves of *Pseuderanthemum carruthersii*: Presence of 3-*O*-Methyl Galactose and Anti-Inflammatory Activity in LPS-Stimulated RAW 264.7 Cells

**DOI:** 10.3390/polym11071219

**Published:** 2019-07-22

**Authors:** Vo Hoai Bac, Berit Smestad Paulsen, Le Van Truong, Andreas Koschella, Tat Cuong Trinh, Christian Winther Wold, Suthajini Yogarajah, Thomas Heinze

**Affiliations:** 1Institute of Biotechnology, Vietnam Academy of Science and Technology, 18 Hoang Quoc Viet, Hanoi, Vietnam; 2Department of Pharmacy, Section of Pharmaceutical Chemistry, University of Oslo, 0316 Oslo, Norway; 3Graduate University of Science and Technology, Vietnam Academy of Science and Technology, 18 Hoang Quoc Viet, Hanoi, Vietnam; 4Friedrich Schiller University of Jena, Institute for Organic Chemistry and Macromolecular Chemistry, Center of Excellence for Polysaccharide Research, Humboldtstrasse, D-07743 Jena, Germany; 5Key Laboratory for Enzyme and Protein Technology, Hanoi University of Science, Hanoi, Vietnam

**Keywords:** *Pseuderanthemum carruthersii*, 3-*O*-methyl galactose, inflammation, MAPK, polysaccharides

## Abstract

*Pseuderanthemum carruthersii* (Seem.) Guillaumin is a native tree in Vietnam. The water extract of the leaves from this tree gives a highly viscous product that has been used to heal wounds and treat inflammations. Our previous studies showed that the leaves of *P. carruthersii* have a high content of polysaccharides. In this study, the structure and influence of the neutral polysaccharide from *Pseuderanthemum carruthersii* (PCA1) on lipopolysaccharide (LPS)-stimulated RAW264.7 cells were investigated. The PCA1 isolated from *P. carruthersii* is a galactan-type polysaccharide, containing galactose (77.0%), 3-*O*-methyl galactose (20.0%), and arabinose (3.0%). Linkage analysis of PCA1 showed that both the 3-*O*-methyl galactose and galactose were 1,4-linked. The presence of 3-*O*-methyl galactose units as part of the polysaccharide is important and can be used as a chemotaxonomic marker. The molecular weight of the PCA1 was 170 kDa. A PCA1 concentration of 30–40 μg/mL strongly inhibited TNFα, IL-1β, and IL-6 inflammatory cytokine production, and reactive oxygen species (ROS) release. PCA1 had inhibitory activities on pro-inflammatory cytokine and ROS release in LPS-stimulated mouse macrophages in vitro through MAPK signaling.

## 1. Introduction

*Pseuderanthemum carruthersii* (Seem.) Guillaumin belongs to the genus *Pseuderanthemum* (family Acanthaceae). In Vietnam, this plant has been used to heal wounds and treat inflammations [1]. The Vietnamese common name is “Xuân hoa đỏ, Ô rô đỏ”. This species grows at many places in Ho Chi Minh and Hanoi city. There are few studies published on the chemical composition and biological effects of this plant. The research on bioactive constituents from *P. carruthersii* has mainly been focused on low-molecular-weight compounds: Two new lignans (pseuderesinol, pseuderanoside) were isolated and identified, as well as a new triterpene (pseuderanic acid) from the dried root of *P. carruthersii*, and pseuderesinol and magnolin inhibited the MCF-7 cancer cell line [2]. Our previous study showed that *P. carruthersii* leaves contain a high concentration of polysaccharides [3].

Today, polysaccharides are widely used in the pharmaceutical industries. For example, cellulose ethers are natural-based polymers applied in personal care products [4]. Pectins [5], beta-glucans [6], and the group of galactomannans [7] have anti-inflammatory activities. They often have an effect on the innate immune system, such as effects on the release of cytokines: Polysaccharides from *Penthorum chinense* Pursh [8], fucan from the algae *Lobophora variegate* reduced the release of TNFα [9] and polysaccharide from the leaves of *Artemisia tripartita* exhibited macrophage-activating activity [10].

Earlier studies have shown that the biological activities of certain plant polysaccharides are related to the occurrence of polysaccharide derivatives with distinct structures. An example is the presence of 3-*O*-methyl galactose in the highly active fractions isolated from *Acanthus ebracteatus* Vahl, which have effects on the complement system [11]. Water-soluble polysaccharides from *Salvia officinalis* L. are composed of 3-*O*-methyl galactose residues. These polysaccharides have immunomodulatory activity [12]. Mannogalactan from *Pleurotus sajor-caju* consists of 3-*O*-methyl galactose, mannose, and galactose, and had anti-inflammatory effects and reduced carrageenan-induced paw edema [13]. Additionally, bioactive heteropolysaccharides from the medicinal fungus *Inonotus obliquus* (Chaga) and *Tinospora sinensis* also contained the uncommon 3-*O*-methyl galactose [14,15].

However, attempts to synthesize a 3-*O*-methyl galactan starting from galactan by an advanced protecting group technique failed due to inherent factors that influenced the selectivity of the methylation, and the biological activity was lower than expected [16]. Research into finding new sources of polysaccharides from plants having bioactivity is needed.

The aim of this paper was to isolate and purify polysaccharides from the leaves of *P. carruthersii*, and determine the monosaccharide composition, analyze the linkage type, and evaluate the anti-inflammatory activity on lipopolysaccharide (LPS)-stimulated RAW264.7 macrophages. The use of this plant has been known in traditional medicine, but the reasons for its particular activity are unknown up to now and must be unraveled. Our results could thus provide a scientific basis for the use of *P. carruthersii* in Vietnamese folk medicine.

## 2. Materials and Methods

### 2.1. Plant Material

Leaves of *Pseuderanthemum carruthersii* (Seem.) Guillaumin (family Acanthaceae) were collected from Hoang Van Thai garden, Thanh Xuan District, Hanoi city, and Dr. Nguyen Thi Thanh Huong, Institute of Ecology and Biological Resources, Vietnam Academy of Science and Technology, Hanoi, Vietnam, identified the sample. Samples were kept in the Department of Plant Biochemistry, Institute of Biotechnology, Vietnam Academy of Science and Technology, Hanoi, Vietnam (sample number: VHB 036). The leaves of *P. carruthersii* were dried at a temperature of 60 °C.

### 2.2. Extraction and Purification of Polysaccharides

The dried leaves from *P. carruthersii* were crushed and extracted with water using a material/solvent ratio of 10 g/250 mL, and the extraction was repeated three times for 5 h at 60 °C. The extracts were centrifuged, and the supernatants were collected and proteins were removed from the polysaccharide solution by treatment with 10% % trichloroacetic acid (TCA) [3]. The Lowry method was used to quantify the protein content, using albumin as the protein standard [17]. After removing protein, the crude polysaccharide from *P. carruthersii* was precipitated with 80% ethanol overnight, followed by centrifugation. The precipitate (50 mg) was dissolved in 5 mL warm water (60 °C) and centrifuged at 10.000 rpm at room temperature (RT), to remove insoluble materials. Five mL of polysaccharide solution was applied onto a Sephadex G-100 gel filtration column (1.5 × 65 cm). Fractions of 5 mL were collected and the carbohydrate content was determined with the phenol–sulfuric acid assay [18]. The polysaccharide fractions from the Sephadex G-100 column were further fractionated using diethylaminoethyl (DEAE) cellulose anion exchange column chromatography. The neutral polysaccharide was eluted with distilled water, followed by elution of the column with a NaCl gradient of 0–2.0 M in deionized water at a flow rate of 1 mL/min. Fractions of 2 mL were collected and the carbohydrate content was determined using the phenol–sulfuric acid assay. The Lowry method was used for checking for the presence or absence of proteins in the collected fractions [17].

### 2.3. Monosaccharide Composition

Procedure a: The monosaccharide composition was determined by gas chromatography (GC) according to Barsett and Paulsen [19]. Purified and dried polysaccharide (1 mg) was weighed into an acid-washed methanolysis glass tube. Mannitol (100 µL) in MeOH (1 mg/mL, internal standard) was added, and then the MeOH was removed with nitrogen. MeOH/HCl (1 mL/3 M) was added to the dried sample. Then, the sample was sealed and placed in an oven at 80 °C for 24 h for methanolysis. The sample was then dried using nitrogen and trimethylsilyl-derivatized prior to analysis by capillary gas chromatography (Thermo Scientific^TM^), as described by Barsett and Paulsen [19].

Procedure b: One mg polysaccharide was hydrolyzed using 2.5 M trifluoroacetic acid (TFA) for 4 h. The TFA was removed, and then the sample was reduced with NaBD_4_ and further treated for production of the alditol acetates, as described in Section 2.4. The sample was analyzed using gas chromatography-mas spectrometry (GC–MS), as described below.

### 2.4. Linkage Analysis by Ethylation

Ethylation of the polysaccharide was basically carried out using the following steps: One mg polysaccharide sample was deprotonated with sodium hydroxide and dimethyl sulphoxide. The polymers were then ethylated using ethyl iodide. Afterwards, the ethylated polysaccharide was hydrolyzed with 2.5 M TFA (trifluoroacetic acid) for 4 h and reduced with NaBD_4_.

Acetic anhydride and 1-methylimidazole were used for acetylation of the partly O-ethylated alditols. Then, O-ethylated alditol acetates were extracted with dichloromethane (DCM) [20]. Finally, the products obtained were analyzed on a GCMS-QP2010 (Shimadzu Corporation, Kyoto, Japan), using a Restek Rxi-5MS silica column (flow was 1 mL/min, temperature: 80–140 °C), as described by Barsett et al. [21]. The GC–MS software, Version 2.10 (Shimadzu Corporation, Kyoto, Japan) was used for analyzing the chromatograms. This procedure was carried out primarily as described by Sims and Bacic [22], apart from replacing methyl iodide with ethyl iodide.

### 2.5. NMR Spectroscopy

The polysaccharide sample (40 mg) was dissolved in 0.55 mL D_2_O with stirring overnight, and sodium trimethylsilylpropionate-*d*_4_ (TSP) was added as an internal reference. The sample was centrifuged prior to the measurement. The spectra were acquired at a temperature of 60 °C on a Bruker Avance 400 NMR spectrometer, operated with resonance frequencies of 400.22 MHz for ^1^H- and 100.65 MHz for ^13^C nuclei. Up to 10 K scans for ^13^C- and 16 scans for ^1^H-NMR spectra were accumulated. In addition to the ^1^H- and ^13^C-NMR measurements, advanced techniques like distortionless enhancement by polarization transfer ^13^C-NMR spectroscopy (DEPT135), ^1^H/^1^H-correlated spectroscopy (^1^H/^1^H-COSY), and heteronuclear single quantum coherence distortionless enhancement by polarization transfer (HSQC-DEPT) measurements were conducted.

### 2.6. Determination of Molecular Weight of Polysaccharides

Size Exclusion Chromatography (SEC) has been used to determinate the molecular weights of polymer samples. A SEC system from JASCO was used including a Degasser DG-2080-53, a pump PU-980, an autosampler AS-2051 Plus, a column oven, a UV-detector UV-975, and a refractive index detector RI-2031 Plus. NaNO_3_ (0.1 M) containing NaN_3_ was used as an eluent at 30 °C. A SUPREMA columns guard/1000/30 Å from Polymer Standards Service was used. Pullulan standards were used for calibration.

### 2.7. Cell Culture

RAW 264.7 murine macrophages from the American Type Culture Collection (ATCC) were maintained in complete medium Dulbecco’s Modified Eagle Medium (DMEM) with 10% Fetal Bovne Serum (FBS), penicillin G (100 IU/mL), and streptomycin (100 μg/mL), sodium pyruvate, and non-essential amino acids.

### 2.8. Cell Viability Assay

Cell viability assessment was performed by using a Cell Counting Kit-8 (CCK-8, Dojindo Laboratories, Kumamoto, Japan) as per the manufacturer’s instructions. Ten μL of CCK-8 solution was incubated for 1 h with polysaccharide-treated cells at 5–40 μg/mL. Absorbance was measured at a wavelength of 450 nm using an enzyme-linked immunosorbent assay (ELISA) reader (Molecular Devices, San Jose, CA, USA). Values from each treatment were calculated as a percentage relative to the untreated matching control (100% survival).

### 2.9. Enzyme-Linked Immunosorbent Assay

RAW 264.7 murine macrophages were treated as indicated and processed for analysis by sandwich ELISA, as previously described [23]. The levels of cytokines (TNF-α, IL-1β, IL-6, and IL-10) were analyzed by an ELISA reagent (BD Pharmingen, San Jose, CA, USA). All assays were performed as recommended by the manufacturers.

### 2.10. Western Blotting Analysis

Specific antibodies against ERK1/2, phospho-(Thr202/Tyr204)-ERK1/2, p38, phospho-(Thr180/Tyr182)-p38, phospho-(Ser180/Ser181)-IκB kinase-α/β, and IκBα were purchased from Cell Signalling (Beverly, MA, USA). Antibody against β-actin was obtained from Santa Cruz Biotechnology (Santa Cruz, CA, USA). After treatment with various concentrations of PCA1 in the presence or absence of 100 ng/mL LPS, the cells were treated as indicated and processed for analysis by western blotting, as previously described [24]. The membranes were developed by a chemiluminescence (ECL) reagents (Amersham-Pharmacia, Little Chalfont, UK).

### 2.11. Measurement of Intracellular Reactive Oxygen Species (ROS)

Intracellular superoxide levels were measured as previously described (Trinh et al., 2018). RAW 264.7 murine macrophages were treated with LPS after incubating with or without PCA1 for 45 min. Then, the cells were incubated with 2 µM dihydroethidium (DHE) (Calbiochem, Merck KGaA, Germany) for 15 min at 37 °C in 5% CO_2_. The cells were examined by laser-scanning confocal microscopy (LSM 510, Zeiss Oberkochen, Germany), and the mean relative fluorescence intensity for each group of cells was assayed by a Carl Zeiss vision system (LSM 510, Zeiss Oberkochen, Germany) and averaged for all groups.

### 2.12. Statistical Analysis

All extractions were performed in triplicate. Data of experiments were presented as mean ± SD (standard deviation) and were analyzed by a Student’s *t*-test according to Bonferroni or ANOVA for multiple comparisons. *p*-values < 0.05 were considered to be significantly different.

## 3. Results and Discussion

### 3.1. Isolation and Purification

The polysaccharides (PCA) were extracted from leaves of *P. carruthersii* with warm water (60 °C) [3]. Our previous studies showed a polysaccharide content in *P. carruthersii* var *atropurpureum* leaves of about 11.6 ± 0.4% in dry weight [3]. The crude polysaccharide was precipitated in ethanol and re-dissolved with warm water, and 10% TCA was used to remove proteins. The supernatant was precipitated with ethanol, collected, washed with ethanol, and dried. For the purification of polysaccharides (PCA), Sephadex G-100 gel filtration chromatography was used to obtain polysaccharide fractions. The elution curve gave a single peak that contains a carbohydrate Figure 1.

The isolated polysaccharide fraction was loaded onto the DEAE–cellulose column. PCA was separated into neutral polysaccharide (PCA1) that was eluted with water, and acidic polysaccharide (PCA2) that was subsequently eluted with NaCl (0.1 M) (Figure 2). PCA1 contained the main part of the polysaccharide present, while sample PCA2 had a carbohydrate content of only 5%. No protein could be detected in these fractions.

### 3.2. Monosaccharide Composition and Molecular Weight of Polysaccharide

Table 1 shows the monosaccharide compositions of PCA1 and PCA2. PCA1 was a neutral polysaccharide containing arabinose (3.0%), galactose (77.0%), and 3-*O*-methyl galactose (20.0%). The GC spectrum is presented in the Appendix A. This trace is the GC trace after methanolysis of PCA1, and the three peaks (13, 14, and 15) represent the trimethylsilyl (TMS) derivatives of the methyl-glycoside of 3-*O*-methyl galactose. This is the first report of the presence of the uncommon 3-*O*-methyl galactose in polysaccharides from leaves of *P. carruthersii.* PCA1 has a high content of galactose (77%). Therefore, PCA1 may belong to the galactan group of polysaccharides. The methanolysis results showed, in addition to the peaks normally obtained for the galactose derivatives, a pattern that we earlier identified as originating from 3-*O*-methyl galactose [11,25]. Previously, 3-*O*-methyl galactose has been found in the genus *Pleurotus* [26,27]. Recently, a polysaccharide was isolated from *Tinospora sinensis*, also containing the composition of 3-*O*-methyl-arabinose, 3-*O*-methyl galactose and galactose with ratio (1.0:6.3:0.9) [15].

The neutral polymer from *Acanthus ebracteatus* was also composed of galactose, 3-*O*-methyl galactose, and arabinose, but in a different ratio (3:4:1) than what was found for PCA1. The high amount of the unusual sugar 3-*O*-methyl galactose may be important as a chemotaxonomic marker, as both *Pseuderanthemum carruthersii* (Seem.) Guillaumin and *Acanthus ebracteatus* Vahl [11] belong to the Acanthaceae family. Our results suggested that PCA1 could be considered as a novel polysaccharide containing this feature.

The SEC curve of sample PCA1 is shown in Figure 3. The symmetrically sharp curve is typical for indicates a single component. Thus, PCA1 is a homogeneous polysaccharide. The average molecular weight of PCA1 was 170 kDa, with a polydispersity of 3.3257. In order to verify the presence of 3-*O*-methyl galactose, the polysaccharide PCA1 was subjected to hydrolysis, reduction, and acetylation, followed by GC–MS. After hydrolysis of the polysaccharide PCA1, the monomers were reduced to the equivalent alditols using sodium borodeuteride as a reducing agent. The deuterium is found on C1, i.e., the position that originally was aldehyde, and fragments containing C1 thus have a mass of one more than those fragments originating from C6 with otherwise similar mass (Figure 4). Thus, fragment 190 shows that the natural methyl group is on Position C3, and this is also verified with fragment 261, a fragment which will contain C6. Cleavage of the carbon chain gives the primary fragments, which are important for the structural elucidation. Fragments originating from cleavage of hexa-acetyl-galactitol will not have these two masses (Figure 4). The total carbohydrate content of PCA2 was only 5%, and this fraction was not studied further. The monosaccharide composition indicated that the polysaccharide part of this fraction was a pectic-type polymer rich in arabinogalactan.

### 3.3. Linkage Analysis of PCA1

The PCA1 fraction was ethylated due to the presence of 3-*O*-methyl galactose in the native polymer. Thus, it was possible to determine the linkages of 3-*O*-methyl galactose. The GC–MS spectrum of PCA1 for identifying 3-*O*-methyl galactose and the types of linkages are presented in the Appendix A. This would not be possible if methylation had been performed. The obtained partly ethylated alditol acetates were analyzed by GC–MS. Based on the data obtained, the results of the mass spectrometry showed that the 3-*O*-methyl galactose was 1,4-linked, and so was the galactose present (Table 2). In addition to these major compounds, traces of ethylated products from terminal arabinofuranose, 1,5-linked arabinofuranose, and the branch point 1,4,6-linked galactose were also observed. The pectic polymer isolated from the bark of the tree *Ulmus glabra* contained 1,4-linked 3-*O*-methyl galactose as well as terminal units [19]. A neutral polymer from *Acanthus ebracteatus* was found to be composed of mainly β-1,4-linked galactose and 3-*O*-methyl galactose [11], and Nagar et al. [15] discovered a polymer from *Tinospora*
*sinensis* composed basically of 1,4-linked 3-*O*-methyl galactose and galactose with 3-*O*-methyl arabinofuranose on Position 6 of 3-*O*-methyl galactose. This latter polymer showed great similarity with the polymer isolated in the present study of *Pseuderanthemum*
*carruthersii.*

### 3.4. Structure Characterization by NMR Spectroscopy

The shape of the NMR spectra is in accordance with the sugar composition of sample PCA1 (Figure 5 and Figure 6).

Polysaccharide-related signals can be found in the range from 3.5 to 5.5 ppm of the ^1^H-NMR spectrum (Figure 5a). The signals at 0 ppm as well as at 0.7 and 2.1 ppm are attributed to the TSP. Moreover, some ethanol remained in the sample, leading to signals at 1.2 and 3.5 ppm. The corresponding resonances of the carbon atoms are located at 19.5 and 55.5 ppm in the ^13^C-NMR spectrum (Figure 5b). At least five different linkage types can be concluded from the NMR data. The main component has a signal for the anomeric position at 4.65/106.92 ppm. Smaller fractions have signals at 4.95/102.51, 5.13/101.69, 5.11/110.20, and 5.29/111.71 ppm. The main components are beta-linked, as concluded from the chemical shift values according to Synytsya and Novak [28]. There are two carbon atoms assigned as methylene groups (60.09 and 63.46 ppm). According to the sugar composition, signals for the methoxy group of 3-*O*-methyl galactose were detected at 3.52/59.59 ppm. The presence of 3-*O*-methyl groups is underpinned by a signal at 3.46/85.64 ppm, which is attributed to Position 3 of the repeating unit that bears a methoxy group. Due to the sugar composition, the peak at 106.92 ppm is C1 of galactose and the peak at 105.9 ppm is C1 of 3-*O*-methyl galactose (Figure 6). The minor peaks in the region from 107 to 110 ppm are attributed to α-l-arabinofuranosides.

The complete peak assignment by combining different NMR methods was hard to achieve. Therefore, the spectra were compared with already published data [11]. Moreover, chemical shifts were calculated using the online tool CASPER (http://www.casper.organ.su.se/casper/, accessed on 25 April 2019).

In this regard, the chemical shifts of the main components could be assigned as follows (^1^H/^13^C chemical shift in ppm, according to reference values given in Table 3). β-(1→4)-linked galactopyranose: Position 1 (4.62–4.66/107.03), Position 2 (3.79/74.60), Position 3 (3.84/76.00), Position 4 (4.18/80.31), Position 5 (3.65-3.80/77.20), and Position 6 (3.65–3.80/63.51). The chemical shifts of the 3-*O*-methylated galactose moieties are quite similar, except the appearance of the signal for Position 3, as discussed above. Signals of lower intensity were tentatively assigned as α-(1→4)-linked galactopyranose: Position 1 (4.90–5.20/101.80–102.77), Position 2 (3.77–4.20/71.40), Position 3 (3.77–4.20/71.40), Position 4 (4.45/80.49), Position 5 (ca. 3.60/75.39), and Position 6 (not assigned/not assigned). Very weak signals could be assigned as Position 1 (110.17 ppm) and Position 5 (68.79 ppm) of α -1→5-linked arabinofuranose.

### 3.5. Anti-Inflammatory Effect of PCA1 on LPS-Induced RAW 264.7 Cells

The cytotoxicity of PCA1 was assessed on the murine RAW 264.7 cell line. The addition of PCA1 to RAW 264.7 macrophage cells did not affect cell proliferation at 5–40 μg/mL (Figure 7). PCA1 had no cytotoxicity on RAW 264.7 cells.

In vitro, LPS-induced RAW 264.7 cells are usually a model for detecting anti-inflammatory effects [8,29]. In response to LPS, macrophages release inflammatory mediators and pro-inflammatory cytokines. LPS-induced RAW 264.7 cells were treated with PCA1 (0–40 μg/mL). A PCA1 concentration of 10 μg/mL significantly inhibited TNFα, IL-1β, and IL-6 inflammatory cytokine production. The release of those cytokines was strongly inhibited at concentrations from 30 to 40 μg/mL (Figure 8). Our results show that PCA1 may attenuate macrophage inflammation by inhibiting LPS-induced pro-inflammatory cytokine secretion.

Plant polysaccharides had macrophage immunomodulation and therapeutic potential [30]. Some polysaccharides, such as polysaccharides from *Penthorum chinense* Pursh or from Cabernet Franc, Cabernet Sauvignon, and Sauvignon Blanc wines, exhibited anti-inflammatory effects in vitro and decreased inflammatory cytokine production. Glucomannan isolated from *Heterodermia obscurata* decreases cytokines IL-1β and TNF-α [8,29,31], and fucan from the algae *Lobophora variegate* reduces the release of TNFα [9].

LPS induces inflammation through the activation of NF-κB and the mitogen-activated protein kinase (MAPK) signaling pathway in macrophages. Furthermore, some studies showed polysaccharide effects on the surface of macrophages via TLR4 [32]. Therefore, we examined the effect of PCA1 on MAPK signaling cascades, including p38 and ERK1/2. RAW 264.7 cells were treated with various concentrations of PCA1 or a solvent control for 45 min before LPS stimulation (100 ng/mL).

The cells were harvested after 30 min of LPS stimulation, lysed with ice-cold lysis buffer, and subjected to Western blot analysis to detect the activation of mitogen-activated protein kinases (MAPKs) (p38 and ERK1/2) and NF-κB.

As shown in Figure 9a, the phosphorylation of ERK1/2 (P-ERK) and p38 (P-p38) were decreased by PCA1 pretreatment. However, PCA1 had no effect on the LPS-induced phosphorylation of NF-κB, as shown Figure 9b. These results indicated that the inhibitory effect of PCA1 on inflammatory cytokines was mediated via the activation of MAPKs by down-regulating the phosphorylation of p38 and ERK ½.

In inflammation, excessive generation of ROS may lead to inflammatory disorders and threatened homeostasis [33].

Lipopolysaccharide (LPS) caused an increase in reactive oxygen species (ROS) production in RAW 264.7 cells (Figure 10). The presence of ROS is indicated with red fluorescence (Figure 11). The treatment with PCA1 at a concentration of 5 µg/mL significantly decreased ROS (reactive oxygen species) levels in macrophage cells (Figure 10). PCA1 with a concentration of 10 µg/mL decreased ROS levels by about 50%, and PCA1 with a concentration of 40 µg/mL decreased ROS levels by about 80%, as compared with the LPS-induced inflammation group without treatment with PCA1 (Figure 10 and Figure 11).

Our results suggest that PCA1 could be an anti-inflammatory agent useful for modulation of LPS-induced inflammatory status through ROS-dependent pathways. Previous studies showed that some polysaccharides belonging to the galactan group have shown anti-inflammatory activity. The exopolysaccharide from *Pleurotus sajor*−*caju* is composed of mannose, galactose, and 3-*O*-methyl galactose, and exhibits anti-inflammation activity [13]. A sulfated polysaccharide from *Agardhiella ramosissima* containing galactose, 3,6-anhydrogalactose, and 6-*O*-methyl galactose shows anti-inflammatory activity as well [34].

## 4. Conclusions

Characterization and bioactivity studies of polysaccharides from *Pseuderanthemum carruthersii* (Seem.) Guillaumin were carried out for the first time. The results showed that the bioactive polysaccharide from *P. carruthersii* (PCA1) belongs to the galactan type, and contains galactose (77.0%), 3-*O*-methyl galactose (20.0%), and arabinose (3.0%). The presence of 3-*O*-methyl galactose is interesting as it has not been found in many polysaccharides. It has been found in two members of the family Acanthaceae, the object of the present study (*Pseuderanthemum carruthersii*), from *Acanthus ebracteatus,* and from *Tinospora sinensis,* as well as in the bark of the elm tree, *Ulmus glabra*. All these plant trees have 3-*O*-methyl galactose present in the polymers as beta-1,4-linked repeating units, while the polysaccharides from the two different fungi mentioned in the introduction both have 3-*O*-methyl galactose as alfa-1,6-linked units in the polymer. The average molecular weight of the PCA1 was 170 kDa. PCA1 isolated from the leaves of the Vietnamese traditional medicinal plant, *Pseuderanthemum carruthersii*, may attenuate macrophage inflammation by inhibiting LPS-induced pro-inflammatory cytokine secretion and ROS release. PCA1 had inhibitory activities on inflammation in LPS-stimulated mouse macrophages in vitro through MAPK signaling. Further research for testing the anti-inflammatory activities on animal models and the other biological effects of PCA1 need to be investigated in the future.

## Figures and Tables

**Figure 1 polymers-11-01219-f001:**
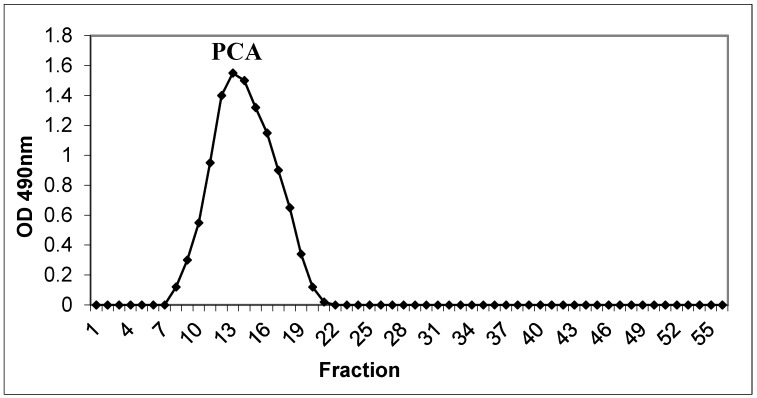
Polysaccharide from *Pseuderanthemum carruthersii* on Sephadex G-100 column chromatography. The column was eluted with water at a flow rate of 0.5 mL/min. A fraction of 5 mL was collected. Elution profiles of polysaccharides: Optical densities (OD) 490 nm.

**Figure 2 polymers-11-01219-f002:**
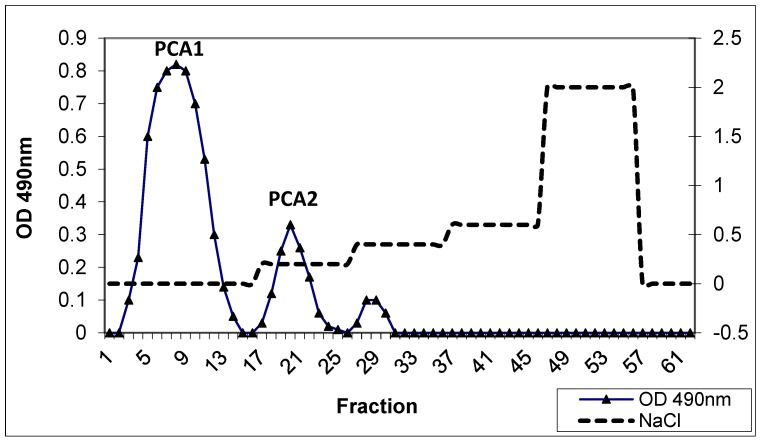
Polysaccharide fractions from *P. carruthersii* eluted using diethylaminoethyl (DEAE) cellulose column chromatography. The neutral polysaccharide was eluted with distilled water, followed by elution of the column with a NaCl gradient (0–2.0 M; 1 mL/min). Fractions of 2 mL were collected. Elution profiles of polysaccharides (OD 490 nm).

**Figure 3 polymers-11-01219-f003:**
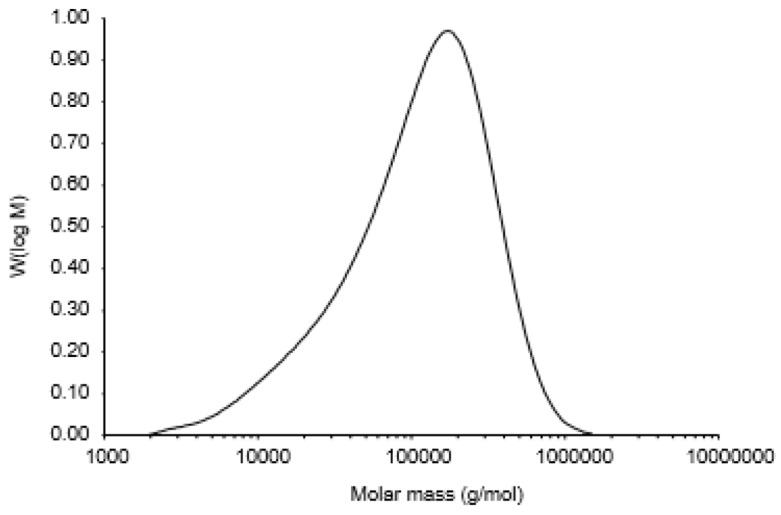
Molecular weight distribution of PCA1, determined by Size Exclusion Chromatography (SEC).

**Figure 4 polymers-11-01219-f004:**
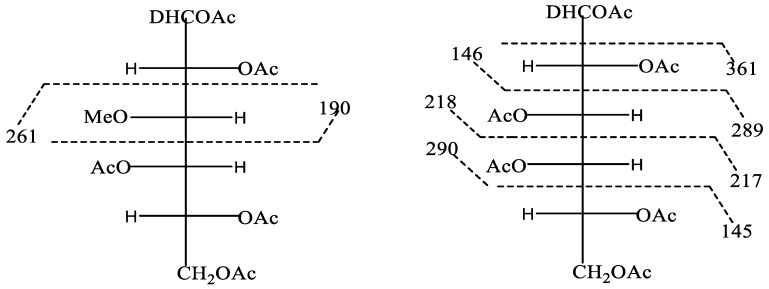
Primary fragments of 3-*O*-methyl pentaacetyl galactitol and hexaacetyl galactitol.

**Figure 5 polymers-11-01219-f005:**
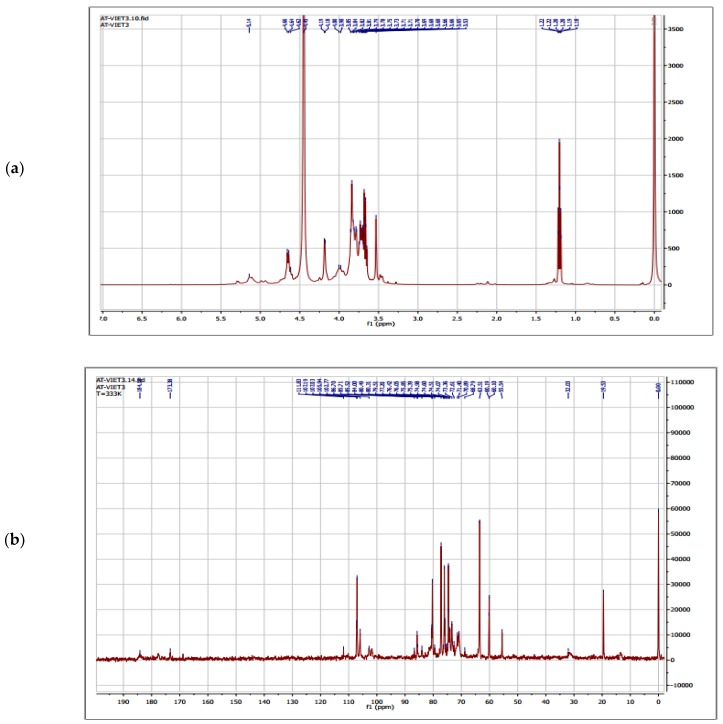
^1^H (**a**) and ^13^C (**b**) NMR spectra of PCA1.

**Figure 6 polymers-11-01219-f006:**
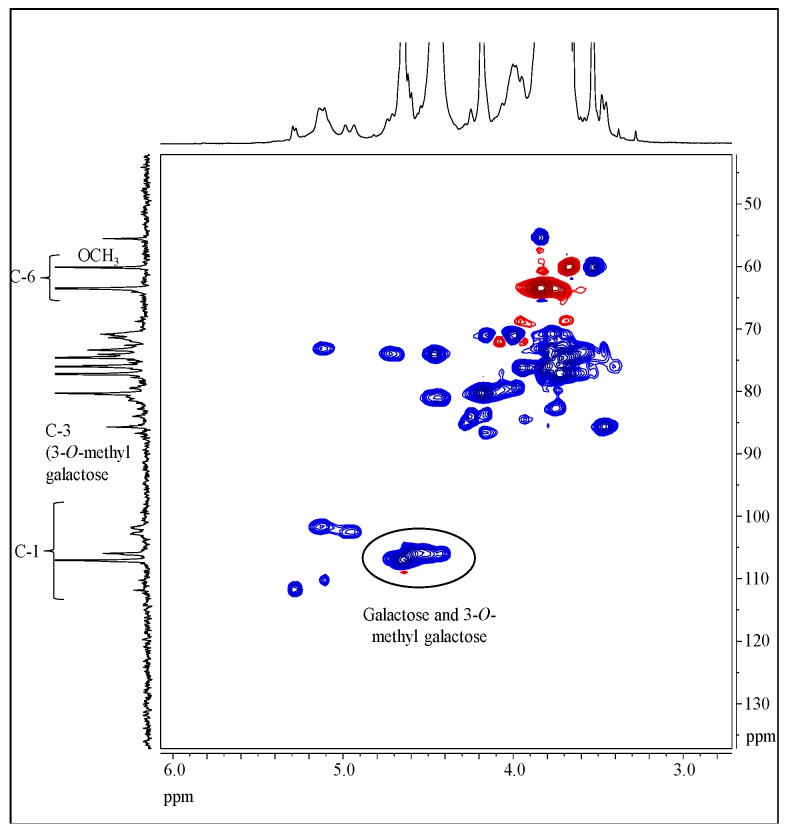
Heteronuclear single quantum coherence distortionless enhancement by polarization transfer (HSQC/DEPT)-NMR spectrum of sample PCA1, recorded in D_2_O.

**Figure 7 polymers-11-01219-f007:**
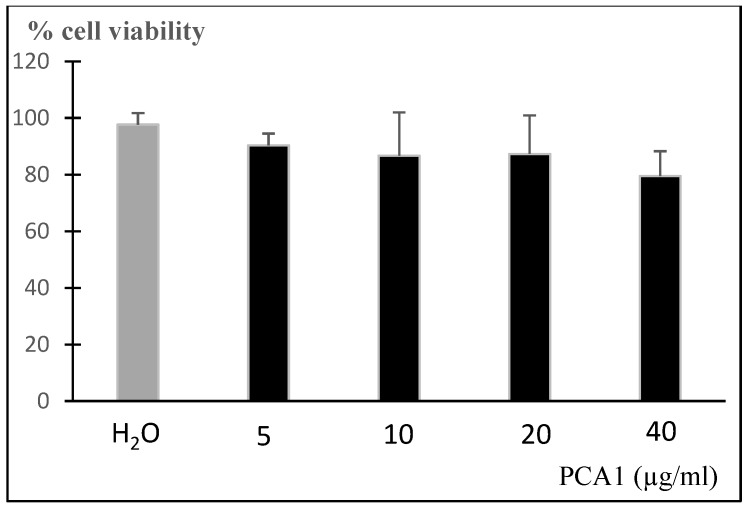
Effect of PCA1 on RAW 264.7 cells. □: Control group (RAW 264.7 cells were pretreated with H_2_O); □: PCA1 treated group (RAW 264.7 cells were treated with PCA1 (5–40 µg/mL).

**Figure 8 polymers-11-01219-f008:**
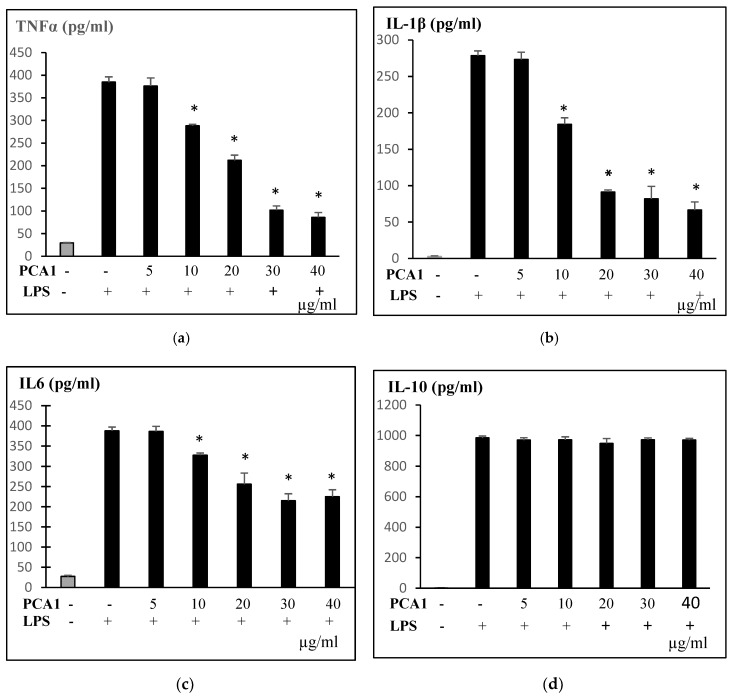
Effect of PCA1 on LPS-induced proinflammatory cytokines TNFα (**a**), IL1β (**b**), TL-6 (**c**), and IL10 (**d**) produced by RAW 264.7 cells. □: Control group (RAW 264.7 cells were pretreated with H_2_O); □: Lipopolysaccharide (LPS) group (RAW 264.7 cells were stimulated with LPS 1 μg/mL and treated with PCA1 (0–40 µg/mL); * *p* < 0.05 compared to LPS group without treatment with PCA1.

**Figure 9 polymers-11-01219-f009:**
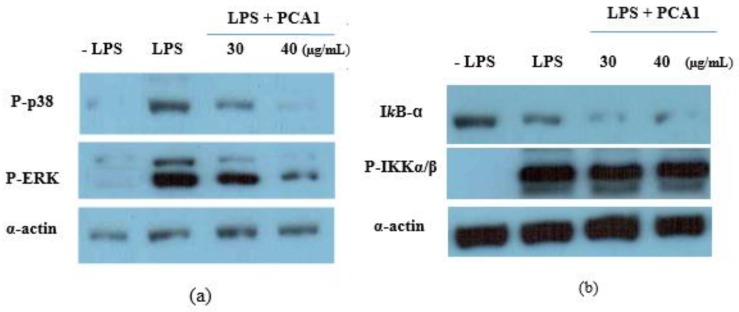
Regulatory effect of PCA1 on the LPS-induced activation of MAPKs (**a**) and NF-κB (**b**) signalling in RAW 264.7. RAW 264.7 cells were treated with concentrations of the solvent control (H_2_O) or PCA1 (30–40 µg/mL) for 45 min before LPS stimulation (1 µg/mL). The cells were harvested after 30 min of LPS stimulation, lysed with ice-cold lysis buffer, and subjected to Western blot analysis to detect the activation of mitogen-activated protein kinases (MAPKs) (p38 and ERK1/2) and NF-κB. The data are representative of three separate experiments.

**Figure 10 polymers-11-01219-f010:**
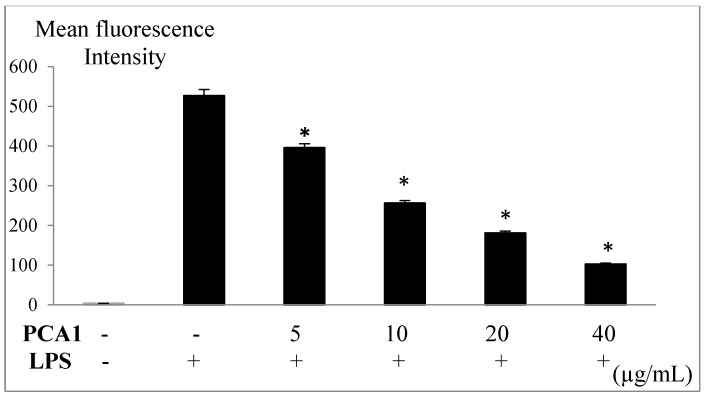
PCA1 inhibited LPS-induced Reactive Oxygen Species (ROS) production in RAW 264.7 cells. □: Control group (RAW 264.7 cells were pretreated with H_2_O). □: LPS group (RAW 264.7 cells were stimulated with LPS 1 μg/mL and treated with PCA1 (0–40 µg/mL). The results are expressed as the mean ± SD of three experiments. * *p* < 0.05 compared to the LPS-induced inflammatory group without treatment with PCA1. PCA1 (0–40 µg/mL) were added to the RAW 264.7 cells before incubating with LPS (bacterial endotoxin 1 μg/mL), and labeled with DHE (dihydroethidium). Then, the fluorescence intensity of the cells was evaluated by laser-scanning microscopy and software system (LSM510).

**Figure 11 polymers-11-01219-f011:**
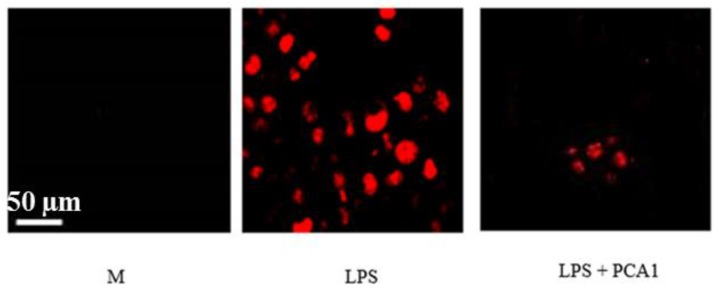
ROS detection in RAW 264.7 cells by laser-scanning microscopy. The RAW 264.7 cells were stimulated with LPS (bacterial endotoxin 1 μg/mL) with and without addition of PCA1 (40 µg/mL), and labeled with DHE (dihydroethidium). Fluorescence was detected with a laser scanning microscope. The presence of ROS is indicated as red fluorescence. M: solvent control (H_2_O) was added to RAW 264.7 cells; LPS: RAW 264.7 cells were stimulated with LPS and treated with H_2_O; LPS+PCA1: RAW 264.7 cells were stimulated with LPS and treated with PCA1 (40 μg/mL).

**Table 1 polymers-11-01219-t001:** Relative monosaccharide composition (Mol %) and *Mw* (Da) of PCA1 and PCA2 fractions from leaves of *P. carruthersii*.

Sugar Composition, Mw	PCA1	PCA2
*Mw* (Da)	1.6826 × 10^5^	-
*Monosaccharide composition* *		
Arabinose	3.0	19.5
Rhamnose		3.7
Fucose		1.8
Xylose		2.3
Galactose	77.0	50.4
Glucuronic acid		2.6
Galacturonic acid		16.1
3-*O*-Methyl galactose	20.0	3.5

*** Mol %: Total carbohydrate content; (-): Not determined.

**Table 2 polymers-11-01219-t002:** Linkage analysis of PCA1 on gas chromatography-mas spectrometry (GC–MS).

Identity, Linkage Type, (Ratio)	Molar Masses of Primary Fragments of the Ethylated Alditol Acetates
1,4 linked 3-*O*-methylgalactose (3,9)	59, 132, 176, 247
1,4 linked galactose (1)	59, 132, 190, 261

**Table 3 polymers-11-01219-t003:** NMR shifts of different structural features.

Backbone	Chemical Shifts (Top: ^13^C, Bottom: ^1^H)							
	1	2	3	4	5	6	6	OCH_3_
→4)β-d-Gal(1→^1^	105.17 4.54	72.92 3.64	74.22 3.72	78.63 4.10	75.42 3.67	61.54 3.66	3.80	
→4)α-d-Gal(1→^1^	101.30 4.98	69.82 3.85	69.69 3.91	79.59 4.07	72.36 4.32	60.93 3.75	3.83	
β-1→4-Galp^2^	105.3	71.6 3.7	74.0 3.8	78.6 4.1	75.5 3.7	61.8–62.2 4.5–4.6	3.9	
3-*O*-Me-Galp^2^	104.5	71.6 3.7	85.64 3.8	78.6 4.1	75.5 3.7	61.8 4.5	62.2 4.6	58.64 3.5
α-1→5- Araf^2^	108.5	n. d.	n. d.	n. d.	68.0			

^1^ CASPER; the peaks in the spectrum appear about 2 ppm higher; ^2^ Carbohydrate Research 339 (2004) 753–762.

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
