# Peer review of "Neutral Polysaccharide from the Leaves of Pseuderanthemum carruthersii: Presence of 3-O-Methyl Galactose and Anti-Inflammatory Activity in LPS-Stimulated RAW 264.7 Cells"

_polymers, 2019, doi:10.3390/polym11071219_

Round 1

Reviewer 1 Report

Neutral polysaccharide from the leaves of  Pseuderanthemum carruthersii: presence of 3-O-methyl  galactose and anti-inflammatory activity in LPS  stimulated RAW 264.7 cells

Vo Hoai Bac, Berit Smestad Paulsen, Le Van Truong, Andreas Koschella, Tat Cuong Trinh, Christian Winther Wold, Suthajini Yogarajah and Thomas Heinze

Herewith I am sending in my reviewer report for the above-mentioned manuscript which is under consideration to be published in polymers.

The article is about a plant called Guillaumin which has traditionally been used for wound healing. The authors investigate the effect of some of its ingredients on LPS challenged cells to determine the anti-inflammatory response. More specifically they were interested in neutral polysaccharide (PCA1) which they isolated from the plant. The authors analysed the compound and investigated its interaction with mouse macrophage cells. I find the question interesting in principle. It is a bit hard to judge for me how much there is already in the literature (Ref 29, is in Vietnamese(?) which I cannot read but the article at least looks different enough). Overall the article is well written and the scientific quality is sufficient.

Line 101: “The  sample was then dried using nitrogen and trimethylsilyl-derivatized prior to analysis by capillary gas chromatography (Thermo Scientific  TM) as described by Barsett and Paulsen” What is the purpose of the trimethylsilyl-derivatization?

Line 143: “treatment were calculated as a percent relative to the” Should be “treatment were calculated as a percentage relative to the”

Line 153: “Antibodie against β-actin was obtained from Santa Cruz” should be “Antibody against β-actin was obtained from Santa Cruz”

Fig 1: In the caption it says “Fragtion numbers”. I believe this should be fraction (also in Fig2)

Line 235: “Thus it was possible to determine the linkages of” there is a comma missing after thus.

Fig 3: there are no numbers at all on the y axis?

Line 298: “LPS-induced RAW 264.7 cells is usually a model for” should be “LPS-induced RAW 264.7 cells are usually a model for”

Caption of Fig 8: the grey and black squares shifted in position

Line 332: “However, PCA1 was not effect on the LPS-induced phosphorylation” should be “However, PCA1 had no effect on the LPS-induced phosphorylation”

Fig 10: the grey/black rectangles got shifted

Fig 11: the caption could be more informative. What is fluorescing here? And what does it mean?

Author Response

Dear Reviewer 1

Thank you very much for your comments. We revised the line and figure according to the your suggestion

Best regards

Vo Hoai Bac 

Reviewer 2 Report

The manuscript entitled "Neutral polysaccharide from the leaves of Pseuderanthemum carruthersii: presence of 3-O-methyl galactose and anti-inflammatory activity in stimulated LPS (RAW 264.7 cells)" is clear and well presented.

However, some minor remarks are to be modified and a major question remains to be addressed.

Concerning the major remark, these are the results of the NMR presented in Table 3 (and unnecessarily repeated in the following paragraph). Thus, it has been repeatedly reported that methylation affects little the chemical displacement of hydrogen on the neighboring carbon, but significantly affects that of Carbon. Also I do not understand that the 13C value of C3 is 74 ppm whether or not O-methylated. I know the complexity of the analysis of these polymers, but it seems to me that it would be necessary on the HSQC to verify that this signal is not near 78 ppm (which will also be verified in HMBC by starting signal H6 which seems singular ).

Regarding minor remarks:
First of all, it is necessary to review the legends of the figures: for example, Figures 10 and 11 do not indicate that the fluorescence is measured on a confocal instrument after DHE treatment, Figure 9 does not indicate a Western-Blot result, etc ...
I do not think either (given the width of the SEC peak that the accuracy of the average value is 10 Da, so the value of 170 kDa seems more appropriate.
I find regrettable the absence of a figure showing the mass spectrum of the "rare" 3-OMe-Gal sugar in the form of alditol and after ethylation of the PS.
Finally, I know that these shortcuts are common in herbal medicine, but even if the anti-inflammatory action on murine macrophage is clearly shown here, it is good (at least in the conclusion) to remember that human immunity is different. In addition, a PS present in the diet or poultice, is probably not present intact in the blood circulating these macrophages. A test on the animal would seem necessary to support this conclusion

Author Response

Dear Reviewer 

Thank you very much for your comments

We revised remarks and corrected our manuscript according to your suggestions.

Best regards

Vo Hoai Bac
